# Effects of deconstructed Tai Chi step training combined with conventional rehabilitation on lower limb function in brunnstrom stage III stroke patients: A randomized controlled trial

Jinxin Chang[1], Pengcheng Qu[2], Yindong Li[2]*

1 School of Physical Education, Shanxi University, Taiyuan, Shanxi Province, China, 2 Chinese Martial Arts Academy, Beijing Sport University, Beijing, China

* liyindong2025@126.com

## Abstract

### Objective

To evaluate whether a deconstructed Tai Chi stepping protocol adapted for patients with Brunnstrom Stage III stroke, when combined with conventional rehabilitation, improves lower limb motor function, walking ability, and joint mobility compared with conventional rehabilitation plus limb synergy training.

### Methods

In this assessor-blinded randomized controlled trial, 52 patients with subacute stroke (Brunnstrom Stage III, ≤ 6 months post-stroke) were randomized in a 1:1 ratio to an experimental group (conventional rehabilitation plus adapted Tai Chi stepping training) or a control group (conventional rehabilitation plus limb synergy training). Both interventions were delivered 5 days per week for 8 weeks. Fifty participants completed the study and were included in the final per-protocol analysis (25 per group). Outcomes were assessed at baseline and after intervention. The primary outcomes were the Fugl-Meyer Assessment for Lower Extremity (FMA-LE) and Holden Walking Function Classification. Secondary outcomes were hip, knee, and ankle range of motion (ROM).

### Results

Both groups improved after treatment, with greater improvement observed in the experimental group. FMA-LE scores increased by 5.6 ± 1.5 points in the experimental group and 1.8 ± 0.8 points in the control group (p < 0.001, Cohen's d = 1.32). In addition, 92% (23/25) of participants in the experimental group achieved Holden Grades II–III compared with 60% (15/25) in the control group (p < 0.001). The experimental group also showed larger gains in joint ROM, including hip flexion (+19.9° vs. +4.2°),

**Data availability statement:** All relevant data are within the manuscript and its Supporting Information files.

**Funding:** The author(s) received no specific funding for this work.

**Competing interests:** The authors have declared that no competing interests exist.

knee external rotation (+9.9° vs. + 3.7°), and ankle dorsiflexion (+7.6° vs. + 1.3°) (all $p < 0.001$).

## Conclusion

Deconstructed Tai Chi stepping training combined with conventional rehabilitation was associated with greater improvement in lower limb motor function, walking ability, and joint mobility than conventional rehabilitation plus limb synergy training in patients with Brunnstrom Stage III stroke. This stage-specific protocol may represent a practical adjunct to stroke rehabilitation, although confirmation in larger trials is still needed.

## Clinical trial registration

International Traditional Medicine Clinical Trial Registry (ITMCTR; a WHO ICTRP Primary Registry), registration number ITMCTR2025000972.

## 1  Introduction

Stroke remains a leading cause of long-term disability worldwide, placing considerable strain on both public health systems and individual well-being due to its high prevalence and the severity of its consequences [1]. Epidemiological evidence shows that 70%–80% of stroke survivors experience some degree of lower limb motor impairment, with Brunnstrom Stage III marking a key transitional period in recovery [2]. At this stage, patients typically present with pronounced extensor spasticity and abnormal gait patterns [3], including hip adduction with internal rotation, knee rigidity, and foot drop accompanied by inversion, all of which severely disrupt walking ability and independence [4]. These impairments reflect not only muscle weakness, but also disrupted neuromuscular coordination, impaired postural control, and limited capacity for effective weight transfer during gait. While existing rehabilitation approaches, such as neurodevelopmental techniques, robotic-assisted therapy, and traditional Chinese acupuncture, can partially address functional deficits, they fall short in two main areas. First, many passive treatment methods rely on specialized equipment and lead to limited patient engagement, which reduces their potential to stimulate neuroplastic changes consistently [5]. Second, these interventions often lack sustained effectiveness over time and fail to adequately target motor impairments specific to Brunnstrom Stage III, such as limited joint synergy and isolation [6]. As a result, there is an urgent need to develop rehabilitation strategies that are safe, effective, and widely applicable in clinical settings, particularly those that prioritize active patient involvement. This need may be especially important during the subacute stage, when motor recovery remains responsive to targeted intervention [7].

In recent years, exercise therapy has emerged as a central element of stroke rehabilitation [8]. A growing body of research supports the use of varied exercise modalities, including aquatic training, progressive resistance exercise, and neuromuscular electrical stimulation, to improve motor performance and life quality by regulating

cardiopulmonary capacity, supporting neuroplastic changes, and coordinating muscle activity [9]. Among these interventions, Tai Chi, a traditional Chinese mind-body practice, has drawn attention for its ability to improve balance, gait symmetry, and lower limb coordination through the integration of mindfulness, breathing, and controlled movement [10]. However, traditional Tai Chi routines are often too complex for individuals in Brunnstrom Stage III, whose motor control limitations may result in compensatory behaviors that intensify spasticity. While earlier studies have tried to adapt Tai Chi or embed it within broader rehabilitation frameworks, a key limitation remains: the lack of Tai Chi step protocols specifically designed for the motor characteristics of Brunnstrom Stage III. Most previously reported modified or simplified Tai Chi interventions were developed for broader stroke populations rather than for patients with marked extensor synergy and poor selective joint control at Stage III. In contrast, a deconstructed Tai Chi protocol is intended not merely to simplify movement difficulty, but to reorganize training around the biomechanical and neurophysiological demands of this stage, including joint isolation, controlled weight shifting, and progressive separation of hip, knee, and ankle movement.These characteristics include the need for isolated control of the hip, knee, and ankle joints, which should be validated using objective measures such as multi-joint range of motion (ROM) tracking [11].

To address this gap, the present study deconstructs the 24-form Tai Chi routine into low-intensity training modules, Zhan Zhuang (standing meditation), Du Li Bu (single-leg stance), and Ce Xing Bu (lateral stepping), specifically adapted for Brunnstrom Stage III patients. In addition, forward and backward stepping components were incorporated to further train postural stability, lower limb coordination, and directional weight transfer. Rather than requiring participants to perform complete Tai Chi forms, the present protocol emphasizes progressive practice of stage-appropriate stepping elements that are more compatible with the movement limitations of Brunnstrom Stage III.These components are integrated with conventional rehabilitation to create a synergy between active and passive therapeutic approaches. The study investigates two core questions: (1) Can the combination of deconstructed Tai Chi step training and conventional rehabilitation significantly improve lower limb motor function and walking ability? (2) Does this combined Tai Chi intervention yield greater multi-directional ROM improvements in the affected hip, knee, and ankle compared to conventional rehabilitation + limb synergy training? We hypothesized that, compared with conventional rehabilitation plus limb synergy training, deconstructed Tai Chi step training combined with conventional rehabilitation would produce greater improvements in lower limb motor function, walking ability, and lower limb joint mobility in patients with Brunnstrom Stage III stroke [12]. Using a randomized controlled trial (RCT) design, we assess the efficacy of the Tai Chi-based protocol through the Fugl-Meyer Assessment for Lower Extremity (FMA-LE), the Holden Walking Function Classification, and a real-time wireless motion capture system. The results aim to refine rehabilitation strategies for Brunnstrom Stage III patients and contribute to the ongoing modernization and clinical validation of traditional exercise practices.

## 2  Study design and methods

### 2.1  Trial design and ethical registration

This parallel randomized controlled trial (RCT) assessed the efficacy of an 8-week deconstructed Tai Chi stepping intervention combined with conventional rehabilitation in improving lower limb function in Brunnstrom Stage III stroke patients. Participants were randomly assigned in a 1:1 ratio to either the Experimental (E) group (n = 26; conventional rehabilitation plus Tai Chi stepping) or the Control (C) group (n = 26; conventional rehabilitation + limb synergy training). Random allocation was performed using a computer-generated random sequence with a block size of 4. The allocation sequence was concealed in sequentially numbered, opaque, sealed envelopes, which were prepared and managed by an independent researcher who was not involved in participant recruitment, intervention delivery, or outcome assessment.

The study received ethical approval from the Ethics Committee of Beijing Sport University (Approval No: 2022093H), and written informed consent was obtained from all participants. Participant recruitment was conducted from September 10, 2022, to November 1, 2022. The intervention and outcome assessments were carried out in the Rehabilitation Department of Heng shui Fifth People's Hospital from November 1, 2022, to January 31, 2023. This trial was registered in the

International Traditional Medicine Clinical Trial Registry (ITMCTR; a WHO ICTRP Primary Registry) under registration number ITMCTR2025000972 Fig 1.

## 2.2 Participant inclusion and exclusion criteria

**2.2.1 Inclusion criteria.** The intervention and outcome assessments were conducted in the Rehabilitation Department of Heng shui Fifth People's Hospital from November 1, 2022, to January 31, 2023. Participants were identified through the hospital's electronic medical records based on a confirmed diagnosis of Brunnstrom Stage III. Sample size was calculated using G*Power 3.1 based on the primary continuous outcome, with $\alpha = 0.05$, power $= 0.80$, and an assumed effect size of $d = 1.2$ derived from the between-group difference reported in Zhao et al [13]. This yielded a required minimum of 20 participants per group. To account for a 20% dropout rate, 52 participants (26 per group) were initially recruited, with 50 completing the trial (25 in each group). Inclusion criteria were: (1) confirmed stroke diagnosis according to the Chinese Guidelines for Acute Ischemic Stroke 2021 or the Intracerebral Hemorrhage Diagnosis and Treatment Guidelines 2019, verified by CT or MRI; (2) lower limb Brunnstrom Stage III (characterized by peak extensor spasticity with hip adduction and internal rotation, knee extension synergy, and active hip/knee flexion in sitting or standing positions without isolated control); (3) age between 30 and 70 years, disease duration ≤6 months, and stable vital signs; (4) absence of severe joint

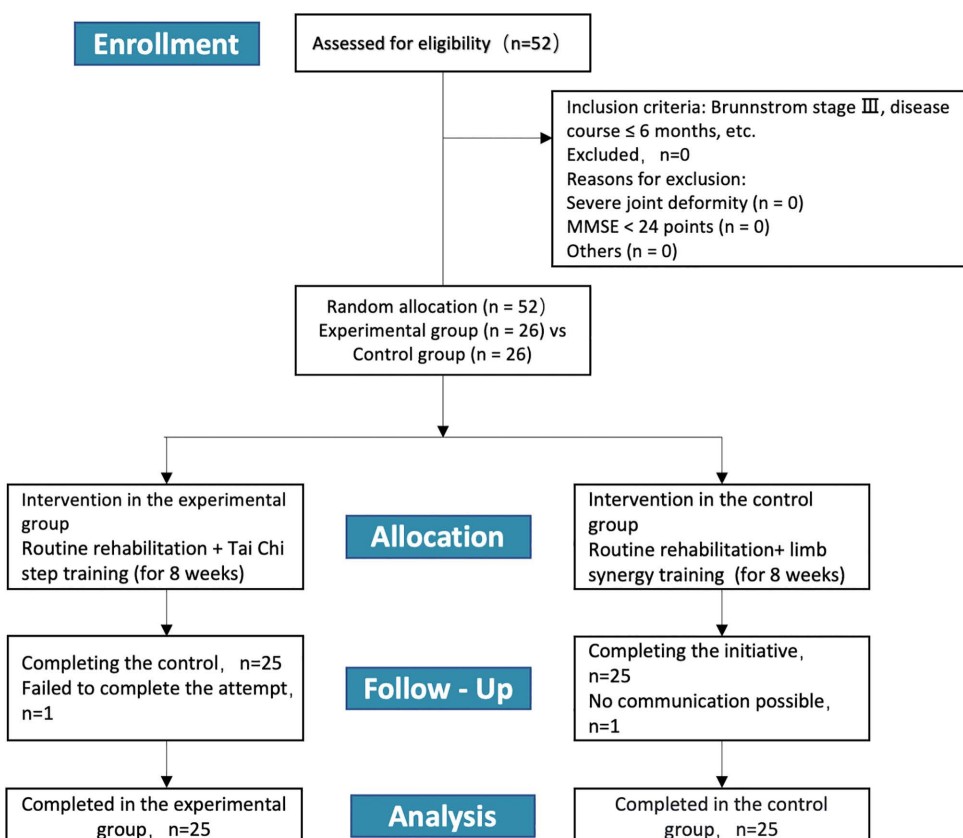

**Fig 1. A CONSORT flowchart: Among 52 patients screened for eligibility (Brunnstrom Stage III, disease duration ≤6 months), none were excluded due to severe joint deformities, Mini-Mental State Examination (MMSE) <24, or other reasons.** After randomization, 26 participants were assigned to each group. One participant from the E group discontinued the intervention, and one from the C group was lost to follow-up, resulting in 25 completers per group for final analysis.

deformity or fracture on the affected side, MMSE score ≥24, and no major cardiopulmonary or neurological comorbidities; (5) voluntary participation with signed informed consent.

**2.2.2 Exclusion, attrition, and termination criteria.** Participants who were found during screening to have previously undisclosed comorbidities, MMSE scores < 24, or other conditions inconsistent with the eligibility criteria were excluded before randomization. After randomization, participants who failed to complete at least 80% of the required sessions (fewer than 4 Tai Chi sessions per week for the E group or fewer than 4 rehabilitation sessions, including limb synergy training, per week for the C group) were regarded as protocol discontinuations rather than eligibility exclusions. One participant in the E group discontinued the intervention at week 4 because of family obligations, and one participant in the C group was lost to follow-up after week 6 and could not be contacted. No other dropouts occurred. Because dropouts occurred after randomization and were not replaced, the final analysis was conducted on a per-protocol basis (n = 25 per group). Intention-to-treat analysis was not performed. Termination criteria included serious adverse events (e.g., joint sprain, fall, or blood pressure ≥ 180/100 mmHg), onset of new severe comorbidities, or recurrence of stroke.

**2.2.3 Blinding and assessor training.** Given the distinctive features of Tai Chi compared with conventional rehabilitation, blinding of participants and therapists was not feasible. To reduce implementation bias, all therapists underwent standardized training, passed competency assessments, and adhered strictly to scripted intervention protocols without providing unscripted cues [14]. Outcome evaluations, including FMA-LE, Holden classification, and ROM, were conducted by two assessors who followed a unified protocol and were blinded to group allocation [15]. To verify the integrity of assessor blinding, after each evaluation session the assessors were asked to guess the participant's group assignment (experimental or control). The blinding was considered successful if the proportion of correct guesses did not exceed chance level (50%). Across all assessments, the assessors' guess accuracy was 38%, suggesting that assessor blinding was likely maintained throughout the study.

## 2.3 Interventions and quality control

**2.3.1 Conventional rehabilitation protocol.** Both groups received standardized neurorehabilitation. Physical therapy included medium-frequency electrical stimulation (20 minutes per session) to activate muscle groups and limb linkage exercises (10 minutes per session) aimed at improving joint mobility [16]. Traditional therapy involved massage (15 minutes per session) to relieve spasticity and acupuncture (15 minutes per session) targeting Zusanli (ST36) and Yanglingquan (GB34) [17]. Sessions were conducted for 60 minutes per day, 5 days per week, over 8 weeks. The conventional rehabilitation protocol was identical in both groups throughout the study period.

**2.3.2 Limb synergy training protocol.** In addition to conventional rehabilitation, the C group received 40-minute limb synergy training sessions designed to address abnormal motor synergy patterns specific to Brunnstrom Stage III, with target heart rate maintained at 40%–60% of maximum heart rate (monitored via Polar sensors, consistent with the E group). Sessions were conducted 5 days per week for 8 weeks, supervised by certified rehabilitation therapists trained in neurodevelopmental techniques.

The protocol included: (1) Slope Standing Training (20 minutes/session, 1 set):Participants stood on a motorized slope board, starting at a 30° inclination and gradually increasing to 60° as tolerated. (2) Limb Coordination Training (10 minutes per set, 2 sets; total 20 minutes/session):Using a four-limb linkage rehabilitation trainer, participants performed bilateral rhythmic movements (flexion/extension, abduction/adduction) under passive-to-active assistive modes, Fig 2. This control intervention was matched to the experimental intervention in session frequency, duration, and therapist supervision.

**2.3.3 Deconstructed Tai Chi stepping protocol.** In addition to conventional rehabilitation, the E group received Tai Chi stepping training adapted from the 24-form Tai Chi routine. This protocol was designed to address the movement characteristics of Brunnstrom Stage III, including extensor synergy dominance, limited selective joint control, and impaired weight shifting. This included *Zhan Zhuang* (standing meditation), *Du Li Bu* (single-leg stance), *Ce Xing Bu* (lateral stepping), and forward/backward stepping. Therapists completed a 2-week training program focused on the motor

**Deconstructed Tai Chi Stepping Protocol**

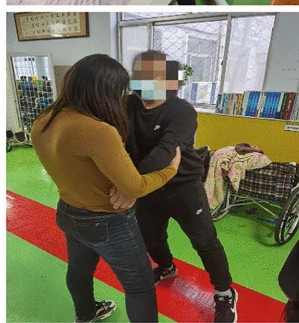
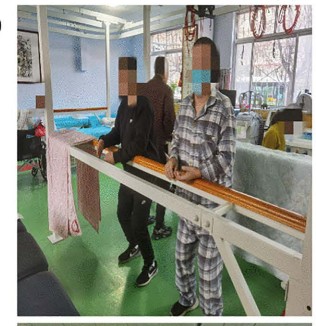
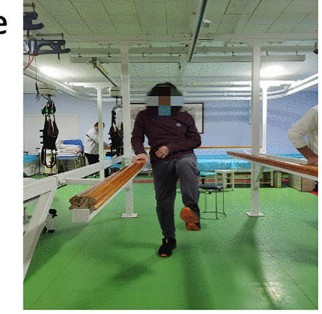
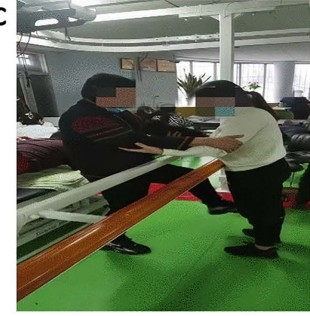
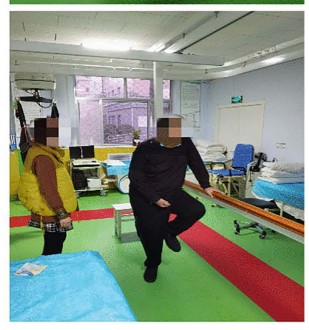

**Limb Synergy Training Protocol**

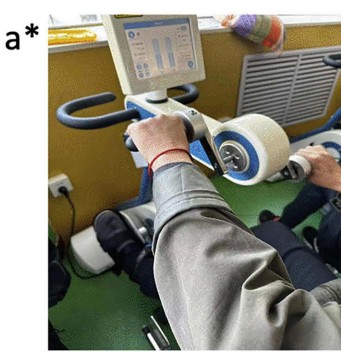
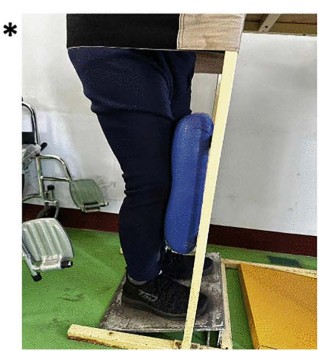

**Fig 2. Intervention design diagram: Figure a: Shows a patient wearing a heart rate monitor.** Figures b to f depict participants in the experimental group performing Zhan Zhuang (standing meditation), Du Li Bu (single-leg stance), Ce Xing Bu (lateral stepping), forward stepping, and backward stepping exercises, respectively. Figures a* and b* illustrate participants in the control group undergoing four-limb linkage practice and slope standing training, respectively.

features of Brunnstrom Stage III, Tai Chi step modifications, and safety guidelines. Therapists were instructed to deliver the intervention according to a standardized movement sequence and safety protocol [18].

The Tai Chi sessions consisted of:

(1) Warm-up (15 minutes): Zhan Zhuang (feet shoulder-width apart, knees flexed at 30°, held for 1.5–3 minutes per set, 2–3 sets) aimed at improving stability and proprioception. Du Li Bu (single-leg knee lift ≥90°, 10 repetitions per set, 2 sets) targeted hip flexion and balance.

(2) Core training (20 minutes): Ce Xing Bu (lateral stepping with 45° hip abduction, 5–8 steps per set, 2–3 sets) emphasized pelvic control. Forward and backward stepping (hip/knee flexion ≥60°, 5 steps per set, 2–3 sets) was used to separate hip-knee synergy [19].

(3) Cool-down (5 minutes): Static stretching of the hamstrings, quadriceps, and tibialis anterior, held for 15–20 seconds per muscle. Exercise intensity was monitored using Polar heart rate sensors, maintaining 40%–60% of the maximum heart rate, calculated as 220 minus age. Each session lasted 40 minutes per day, 5 days per week, for 8 weeks, and was supervised by certified Tai Chi instructors with rehabilitation experience [20].

**2.3.4 Quality control and safety monitoring.** Tai Chi movements followed the standards of the 24-Form Tai Chi Competition Routine, with individualized modifications based on each patient's functional level. Safety procedures included blood pressure checks before each session (only participants with readings below 160/100 mmHg were cleared to proceed) and real-time monitoring of heart rate and respiratory rate. Attendance and heart rate data were recorded to track compliance, with a minimum of 80% session completion required to ensure consistency of the intervention.

To support intervention fidelity and methodological rigor, additional quality control procedures were implemented throughout the intervention period. Adherence to the intervention was verified daily using attendance logs. Participants who completed fewer than 80% of the scheduled sessions were considered non-adherent according to the predefined protocol. Bias minimization measures included standardized therapist training, scripted intervention procedures without unscripted cues, and blinded outcome assessment, the integrity of which was evaluated as described in Section 2.2.3.

## 2.4 Outcome measures

Outcome measures were assessed at baseline before randomization and again immediately after completion of the 8-week intervention period. The primary outcomes of this study were lower limb motor function and walking ability, assessed using the Fugl-Meyer Assessment for Lower Extremity (FMA-LE) and the Holden Walking Function Classification, respectively. The secondary outcomes were hip, knee, and ankle joint range of motion (ROM).

(1) FMA-LE: Used to assess lower limb motor function, with a total possible score of 34 points.

(2) Holden Walking Function Classification: Evaluated walking independence and safety across six levels, based on observations during level walking, incline/decline walking, turning, and the need for balance support. This study focused on Grades I–III to reflect the functional status and goals relevant to Brunnstrom Stage III patients.

(3) Joint ROM: Assessed using a real-time wireless motion capture system (FAB) in combination with goniometry. Each joint was measured three times, with the maximum value recorded. Hip ROM included flexion, extension, and abduction; knee ROM included flexion, external rotation, and internal rotation; ankle ROM included dorsiflexion, plantarflexion, inversion, and eversion.

## 2.5 Statistical analysis

All data were processed using SPSS 26.0. Continuous variables are presented as mean ± standard deviation (SD), while categorical variables are presented as frequency (%). Baseline characteristics were summarized descriptively because of the randomized study design. For continuous outcomes, a 2 × 2 repeated-measures ANOVA (group × time) was performed for FMA-LE and joint ROM outcomes to assess time effects, group effects, and group × time interaction effects. When appropriate, between-group comparisons of change scores (post-intervention minus baseline) were additionally examined using independent t-tests for normally distributed variables and Mann-Whitney U tests for non-normally distributed variables. Pre- and post-intervention comparisons within groups were analyzed using paired t-tests as supplementary analyses. Holden Walking Function Classification, as an ordinal outcome, was analyzed using the Mann-Whitney U test. Effect sizes are reported as Cohen's d for t-tests and partial eta squared ($\eta^2$) for ANOVA results, each accompanied by 95% confidence intervals where appropriate. No formal adjustment for multiple comparisons was applied; therefore, secondary ROM analyses should be interpreted cautiously. A two-sided $p < 0.05$ was considered statistically significant. All analyses

were conducted by an independent statistician to ensure objectivity and reproducibility. The primary analysis was conducted on a per-protocol basis (n = 25 per group), as described in Section 2.2.2.

## 3 Results

### 3.1 Baseline demographic and clinical characteristics

Each group included 25 participants. Baseline variables, age, sex, disease duration, BMI, stroke type (ischemic/hemorrhagic), affected side (left/right), presence of hypertension and diabetes, Barthel Index, and Berg Balance Scale scores, were compared between the E and C groups. Baseline demographic and clinical characteristics were generally comparable between the two groups at study entry, as shown descriptively in Table 1.

### 3.2 Fugl-Meyer assessment for lower extremity (FMA-LE)

No significant difference in FMA-LE scores was observed at baseline between groups (E group: 10.2 ± 1.1; C group: 10.3 ± 1.0). A 2 × 2 repeated-measures ANOVA (group × time) revealed a significant interaction effect (F (1,48) = 28.6, $p < 0.001$, partial η² = 0.37), indicating that the improvement over time differed between groups. After the intervention, the E group achieved a significantly greater improvement (Δ = +5.6 ± 1.5, t = 12.741) than the C group (Δ = +1.8 ± 0.8, t = 6.325). The between-group difference in change scores was 3.8 points (95% CI: 2.1 to 5.5), which exceeded the minimal clinically important difference (MCID) for post-stroke lower limb function (reported as 2.5 points), indicating clinical relevance. The effect size was large (Cohen's d = 1.32, 95% CI: 0.89 to 1.75; partial η² = 0.21). The between-group difference was statistically significant ($p < 0.001$). Results are summarized in Table 2.

### 3.3 Holden walking function classification

At baseline, all participants in both groups were classified as Holden Grade I. Following the intervention, the E group had a higher proportion of participants reaching Grades II or III (92%, 23/25) than the C group (60%, 15/25). The risk ratio was 1.53 (95% CI: 1.21–1.93), and the odds ratio was 7.67 (95% CI: 2.14–27.45), both indicating statistical significance ($p < 0.001$) (Table 3).

### 3.4 Hip, knee, and ankle joint range of motion (ROM)

All ROM results are presented as mean ± SD. For each ROM outcome, a 2 × 2 repeated-measures ANOVA (group × time) was performed. Significant group × time interactions were found for all ROM measures (p < 0.01 for all), confirming that the

**Table 1. Comparison of baseline characteristics between the E group and C group.**

| Characteristic | E Group (n = 25) | C Group (n = 25) |
|---|---|---|
| Age (years) | 56.2 ± 9.8 | 58.3 ± 10.5 |
| Sex (male/female) | 18/7 | 16/9 |
| Disease duration (months) | 2.5 ± 1.2 | 2.8 ± 1.4 |
| BMI (kg/m²) | 24.50 ± 3.20 | 25.10 ± 3.50 |
| Stroke type (ischemic/hemorrhagic) | 19/6 | 21/4 |
| Affected side (left/right) | 13/12 | 14/11 |
| Hypertension | 18 | 20 |
| Diabetes | 9 | 7 |
| Barthel Index (score) | 45.6 ± 8.2 | 43.9 ± 7.8 |
| Berg Balance Scale (score) | 28.3 ± 5.1 | 27.8 ± 4.9 |

Note: Data are presented as mean ± SD or n. Baseline characteristics are presented descriptively because of the randomized study design.

**Table 2. FMA-LE scores before and after intervention.**

| Group | Pre-intervention | Post-intervention | Change (Δ) | 95% CI of Δ | p-value (between) |
|---|---|---|---|---|---|
| E Group (n=25) | 10.2±1.1 | 15.8±2.3 | +5.6±1.5 | (4.9, 6.3) | <0.001 |
| C Group (n=25) | 10.3±1.0 | 12.1±1.9 | +1.8±0.8 | (1.4, 2.2) | |

Note: Between-group comparison of change scores was performed using independent t-test. Cohen's d=1.32 (95% CI: 0.89–1.75), η²=0.21.

**Table 3. Comparison of holden walking function classification between groups.**

| Grade | E Group (n=25) | C Group (n=25) |
|---|---|---|
| I (pre) | 25 | 25 |
| I (post) | 2 | 10 |
| II (post) | 15 | 12 |
| III (post) | 8 | 3 |

Note: Holden classification: Grade I (fully dependent), Grade II (substantial assistance), Grade III (moderate assistance). Post-intervention, 92% of E group achieved Grade II or III vs. 60% of C group (Z=3.891, p<0.001).

intervention effects differed between groups Between-group comparisons of change scores were performed using independent t-tests, with 95% confidence intervals for the mean differences.

**3.4.1 Hip joint ROM.** The E group showed significantly greater gains in hip flexion (Δ=+19.9°, 95% CI: 16.2° to 23.6°), extension (Δ=+9.0°, 95% CI: 6.5° to 11.5°), and abduction (Δ=+8.8°, 95% CI: 6.1° to 11.5°) than the C group (flexion: Δ=+4.2°, 95% CI: 2.0° to 6.4°; extension: Δ=+1.6°, 95% CI: −0.2° to 3.4°; abduction: Δ=+3.5°, 95% CI: 1.2° to 5.8°). All between-group differences were significant (p<0.01) (Fig 3).

**3.4.2 Knee joint ROM.** The E group achieved more pronounced improvements in knee flexion (Δ=+9.7°, 95% CI: 7.8° to 11.6°), external rotation (Δ=+9.9°, 95% CI: 7.5° to 12.3°), and internal rotation (Δ=+8.8°, 95% CI: 6.4° to 11.2°) compared to the C group (flexion: Δ=+1.4°, 95% CI: −0.5° to 3.3°; external rotation: Δ=+3.7°, 95% CI: 1.8° to 5.6°; internal rotation: Δ=+3.0°, 95% CI: 1.0° to 5.0°). Between-group differences were statistically significant (p<0.05 to p<0.001) (Fig 4).

**3.4.3 Ankle joint ROM.** The E group showed substantial gains in ankle dorsiflexion (Δ=+7.6°, 95% CI: 6.0° to 9.2°), plantarflexion (Δ=+12.7°, 95% CI: 10.2° to 15.2°), inversion (Δ=+10.3°, 95% CI: 8.1° to 12.5°), and eversion (Δ=+5.9°, 95% CI: 4.2° to 7.6°). These improvements significantly exceeded those observed in the C group, which showed increases of dorsiflexion (Δ=+1.3°, 95% CI: 0.5° to 2.1°), plantarflexion (Δ=+3.1°, 95% CI: 1.8° to 4.4°), inversion (Δ=+2.5°, 95% CI: 0.9° to 4.1°), and eversion (Δ=+1.8°, 95% CI: 0.6° to 3.0°). All between-group differences were significant (p<0.001 for all, except eversion p=0.005) (Fig 5).

## 4 Discussion

This randomized controlled trial found that deconstructed Tai Chi stepping combined with conventional rehabilitation was associated with greater improvement in lower limb motor function, walking ability, and multi-joint mobility than conventional rehabilitation plus limb synergy training in patients with Brunnstrom Stage III stroke. The E group demonstrated greater progress across key outcomes, including FMA-LE scores (Δ=+5.6 vs. Δ=+1.8 in the C group), a higher percentage of patients reaching Holden Grades II or III (92% vs. 60%), and more pronounced improvements in ROM at the hip, knee, and ankle. Taken together, these findings suggest that a stage-specific Tai Chi stepping protocol may provide a useful adjunct to routine neurorehabilitation, particularly when the intervention is designed to address the abnormal extensor

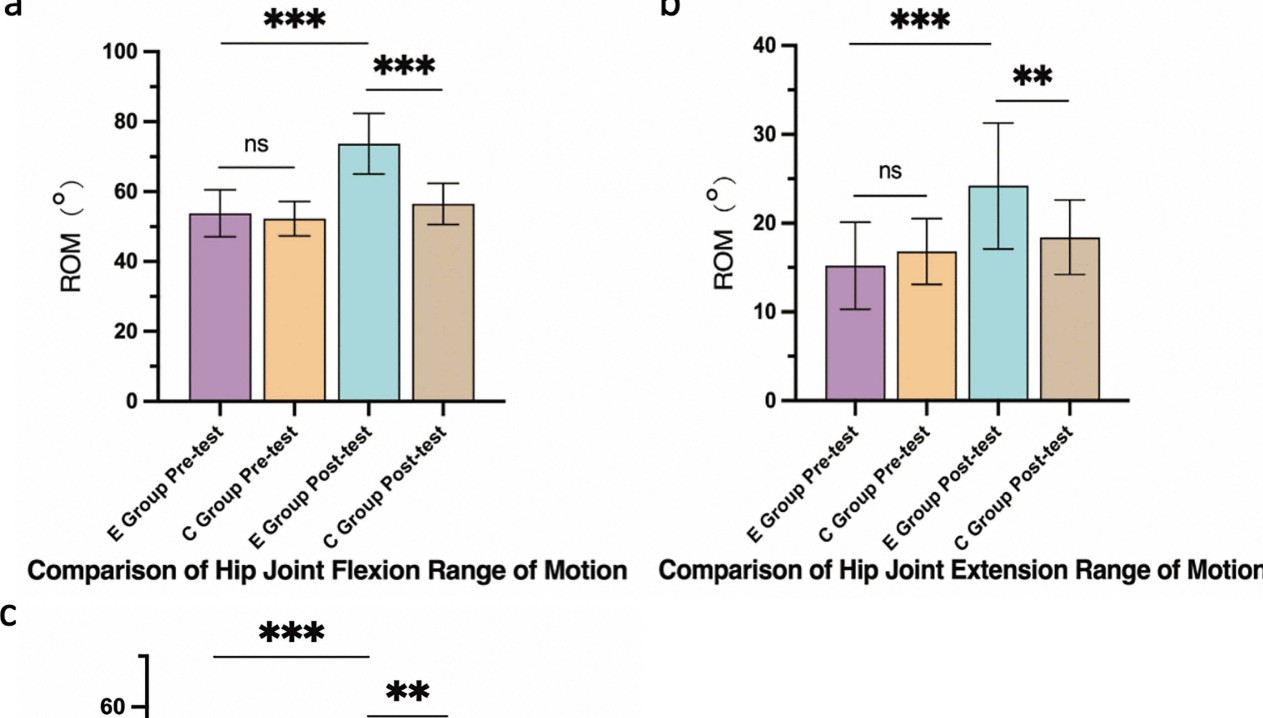

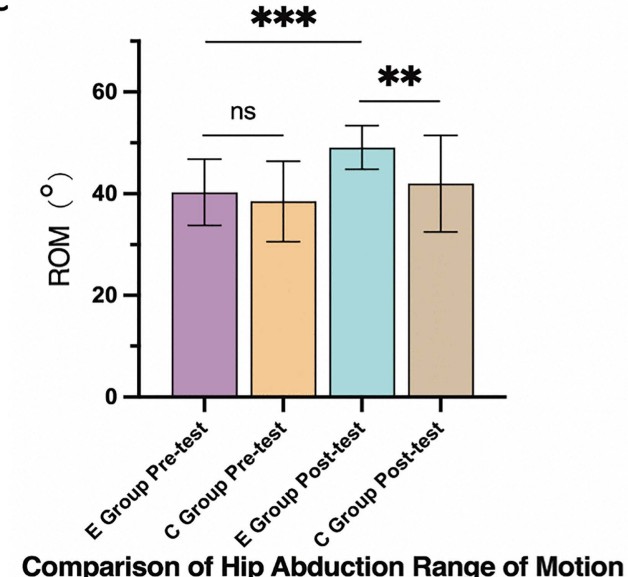

**Fig 3. Hip joint ROM improvements. (a)** Flexion, (b) extension, (c) abduction. Data are mean±SD. ***$p<0.001$, **$p<0.01$, *$p<0.05$, ns: not significant.

synergy and limited selective control characteristic of Brunnstrom Stage III. Rather than interpreting these findings as evidence for broad superiority of Tai Chi in all stroke settings, the present results are more appropriately understood as support for a targeted, stage-adapted movement strategy within a defined rehabilitation context.

### 4.1 Neuromuscular mechanisms underlying Tai Chi step training

The therapeutic effects of Tai Chi step training can be explained through three interconnected mechanisms: neural remodeling, reorganization of muscle synergy, and enhancement of proprioception. From the perspective of motor control theory

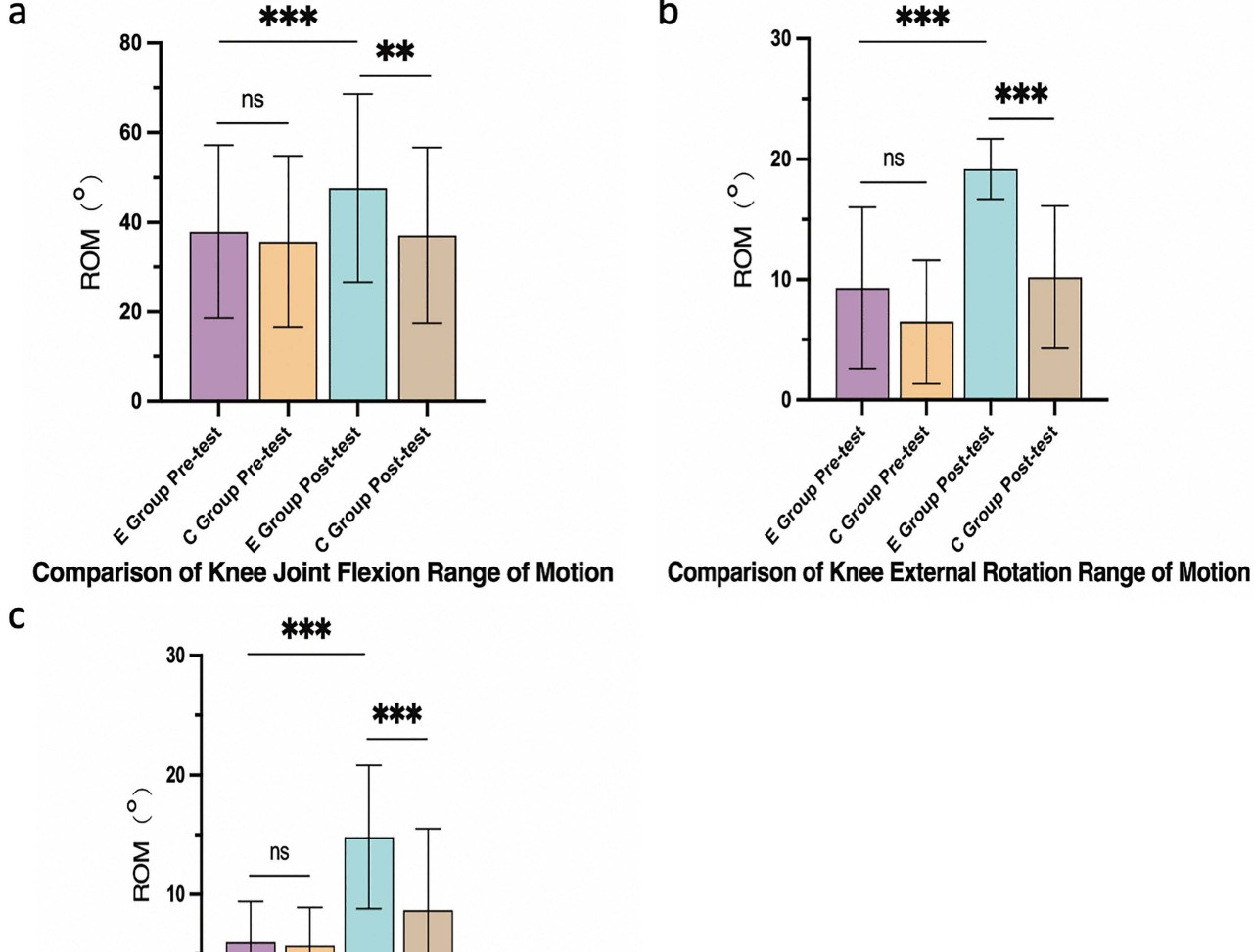

**Fig 4. Knee joint range of motion (ROM) improvements. (a)** Flexion, **(b)** external rotation, **(c)** internal rotation. Data are mean±SD. ***p < 0.001, **p < 0.01, *p < 0.05, ns: not significant.

and neurorehabilitation, the present protocol emphasizes task-specific repetition, controlled weight transfer, postural alignment, and progressive joint dissociation rather than practice of complete Tai Chi routines. This distinction is particularly important in Brunnstrom Stage III, in which abnormal extensor synergy and impaired selective control often constrain efficient gait-related movement.

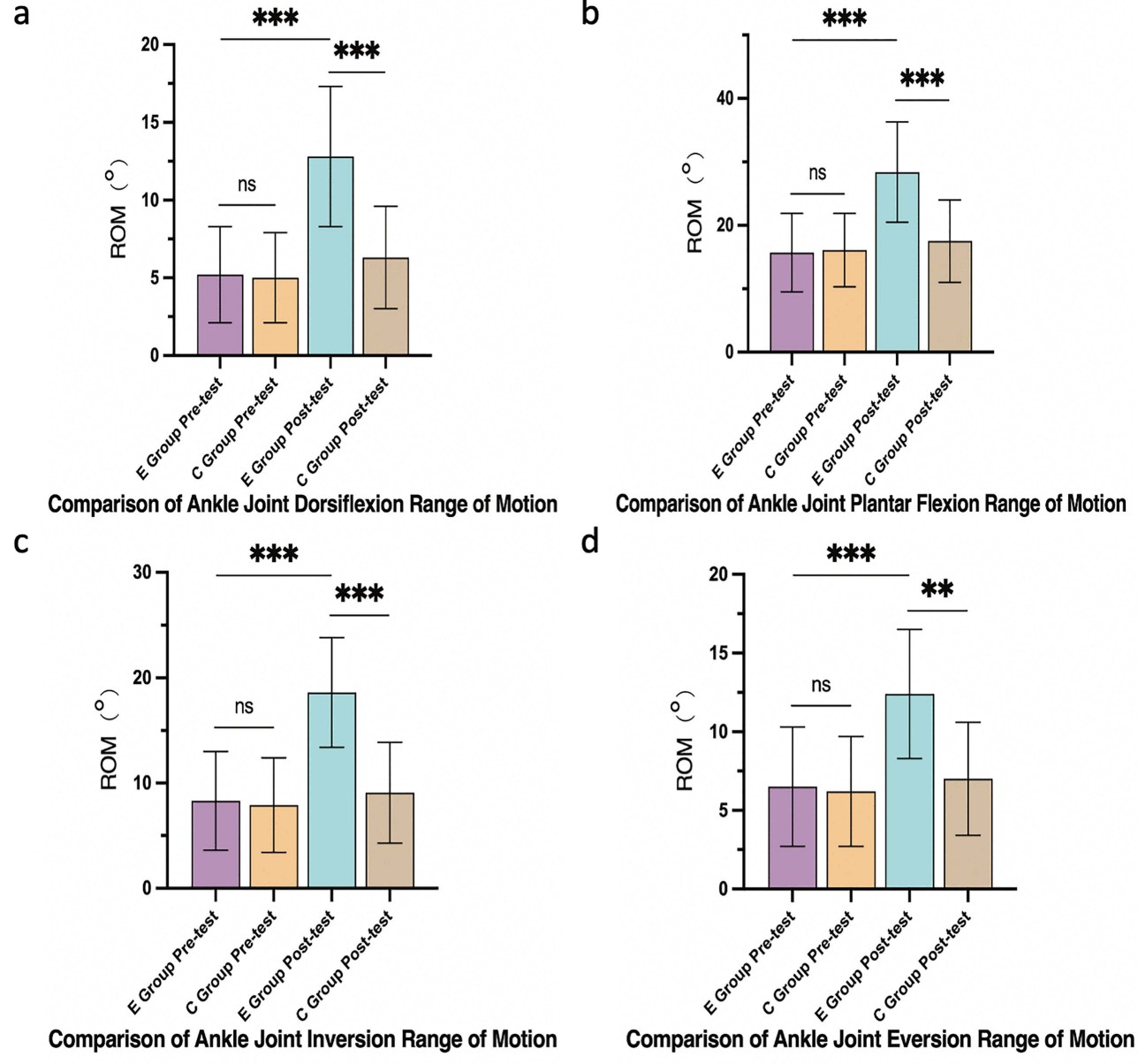

**Fig 5. Ankle joint ROM improvements.** (a) Dorsiflexion, (b) plantarflexion, (c) inversion, (d) eversion. Data are mean±SD. ***$p<0.001$, **$p<0.01$, *$p<0.05$, ns: not significant.

(1) Central Neural Plasticity: Drawing from motor control theory, Tai Chi's emphasis on intentional movement and rhythmic control may reinforce connections between the sensorimotor cortex and cerebellum, promoting neural adaptation [21]. The substantial FMA-LE improvement (Δ=+5.6) supports existing meta-analytical evidence for Tai Chi's effect on motor recovery (*Jiang et al., 2022*) [22]. These changes likely result from improved proprioceptive input during movement execution, which suppresses maladaptive patterns such as hip adduction and foot drop by reshaping central motor output [23]. Although no neurophysiological measures were collected in the present study, this interpretation is broadly consistent with prior Tai Chi-related research suggesting that structured movement practice can facilitate

sensorimotor integration and adaptive neural reorganization [24]. In this sense, the present findings fit within a motor relearning framework rather than a purely symptomatic explanation of spasticity reduction.

(2) Muscle Synergy Reorganization: In Brunnstrom Stage III, extensor-dominant synergy often arises from spasticity. Tai Chi components like *Zhan Zhuang* (standing meditation) and *Du Li Bu* (single-leg stance) engage antagonist muscles such as the hamstrings and tibialis anterior, offsetting excessive quadriceps activity and contributing to more balanced muscular activation (*Sherrington et al., 2019*) [25]. Gains in knee external rotation (Δ = +9.9°) and ankle dorsiflexion (Δ = +7.6°) indicate better joint separation, which may reflect improvement in selective motor control and a reduction in maladaptive co-contraction. This interpretation is supported by previous EMG-based and mechanistic studies suggesting that Tai Chi-related training may improve neuromuscular coordination and movement efficiency [26].

(3) Proprioception and Dynamic Balance: Movements such as *Ce Xing Bu* (lateral stepping) and weight-shift drills challenge pelvic control and stimulate spinal-cerebellar proprioceptive circuits, refining postural responses during gait [27]. The E group's higher rate of achieving Holden Function Grades II and III (92%) points to Tai Chi's role in supporting walking independence, echoing outcomes associated with core stability-focused gait rehabilitation [28]. More broadly, structured exercise may contribute not only to functional improvement but also to neurobiological recovery processes relevant to stroke rehabilitation. In this sense, the present intervention may have promoted recovery through both biomechanical practice and repeated sensorimotor engagement.

## 4.2 Synergistic effects of combined interventions

Conventional rehabilitation techniques such as massage and electrotherapy primarily reduce spasticity through passive means, while Tai Chi step training emphasizes active joint isolation and proprioceptive engagement. This dual approach—passive spasticity reduction plus active motor relearning—produced synergistic improvements in the E group, as evidenced by larger FMA-LE gains (Δ = +5.6 vs. C group's +1.8) and higher Holden II/III rates (92% vs. 60%). [29]. In contrast, the C group's limb synergy training, while promoting bilateral movement coordination, lacked the joint-specific focus needed to address Brunnstrom III's extensor synergy and limited ROM. For example, knee external rotation improved by +9.9° in the E group versus +3.7° in the C group, reflecting Tai Chi's unique ability to activate antagonist muscles and suppress co-contraction [30].

The between-group difference in FMA-LE change scores also appears potentially clinically relevant. However, interpretation against minimal clinically important difference thresholds should remain cautious, because reported MCID values for lower extremity outcomes may vary according to stroke stage, baseline severity, and the measurement context. Therefore, the present findings are more appropriately interpreted as indicating a potentially meaningful functional advantage rather than definitive proof of clinical benefit in all stroke populations.

## 4.3 Clinical innovations and implications

This study addresses a major gap in rehabilitation for Brunnstrom Stage III patients [31]. Unlike Tai Chi protocols tailored to Stages IV–V, the stepwise breakdown used here produced larger improvements in hip abduction (Δ = +8.8° vs. C group's Δ = +3.5°, $p = 0.013$) and ankle dorsiflexion (Δ = +7.6° vs. Δ = +1.3°, $p < 0.001$). These results likely reflect the protocol's focus on peak spasticity and joint limitations, with movements like *Zhan Zhuang* and *Ce Xing Bu* specifically activating the hip abductors and ankle dorsiflexors. This targeted strategy supports early intervention during a period of increased neuroplasticity following stroke.

An important clinical feature of the present protocol is that it does not simply reduce the complexity of Tai Chi for general stroke survivors; rather, it reorganizes selected stepping elements to match the motor constraints of Brunnstrom Stage III. In this respect, its clinical value may lie less in its cultural origin and more in its compatibility with stage-specific rehabilitation principles such as joint dissociation, controlled loading, and task-oriented balance training [32].

Compared with advanced technologies such as robotic gait training, this Tai Chi-based protocol offers distinct benefits [33]: low cost, minimal equipment needs, and adaptability to community or home-based use [34]. The E group's 92% attainment of Holden Function Grade II or III demonstrates its practicality in settings with limited rehabilitation resources, addressing global disparities in access to post-stroke care [35]. This also raises the possibility that the protocol could be adapted beyond East Asian cultural settings, provided that its core training elements are translated into standardized rehabilitation language and delivered with appropriate therapist guidance.

### 4.4 Limitations and future directions

Several limitations should be noted. First, the relatively small sample size (n = 25 per group) and the inclusion of only subacute-phase patients (disease duration ≤6 months) limit the scope of these findings. Larger studies with longer follow-up periods are needed to evaluate long-term outcomes and applicability to chronic cases. Second, the lack of neurophysiological assessments such as fMRI or surface electromyography restricts our understanding of central-peripheral neuromuscular interactions. Third, the impossibility of blinding participants and therapists may introduce performance bias, highlighting the need for objective outcome measures such as gait kinematics in future trials. Lastly, the hospital-based delivery model reduces accessibility; developing home-based formats could increase adherence and broaden the reach of this intervention.

Additional limitations should also be acknowledged. Because the present analysis was conducted on a per-protocol basis after post-randomization dropout, some attrition-related bias cannot be excluded. Furthermore, the potential influence of sex distribution, age-related variability, and stroke subtype on treatment response was not explored in the current sample. The analysis of multiple secondary ROM outcomes may also have increased the risk of type I error, particularly if no formal correction for multiple comparisons was applied.

Future studies should therefore pursue multicenter validation in larger and more diverse populations, incorporate longer-term follow-up, and examine whether this stage-specific Tai Chi-based model can be transferred effectively to community and home-based settings outside East Asia. In addition, wearable motion sensors, kinematic monitoring, and surface EMG could be used to improve mechanistic interpretation, reproducibility, and real-world tracking of movement quality during rehabilitation.

## 5 Conclusion

In patients with Brunnstrom Stage III stroke, deconstructed Tai Chi stepping training combined with conventional rehabilitation was associated with greater improvement in lower limb motor function, walking ability, and joint mobility than conventional rehabilitation plus limb synergy training. The intervention's focus on joint isolation, weight shifting, and coordinated lower limb control appears to be compatible with the movement characteristics of this recovery stage. These findings suggest that a stage-specific deconstructed Tai Chi protocol may represent a practical adjunct to stroke rehabilitation. However, given the modest sample size, per-protocol analysis, and lack of long-term follow-up, further multicenter studies are needed before broader clinical application can be recommended.

### Supporting information

**S1 File. CONSORT_2025_editable_checklist-2.**
(DOCX)

**S2 File. Trial research protocol translated version.**
(DOCX)

### Author contributions

**Conceptualization:** Pengcheng Qu.

**Data curation:** Pengcheng Qu.

Formal analysis: Pengcheng Qu.

Investigation: Jinxin Chang.

Methodology: Jinxin Chang.

Project administration: Yindong Li.

Supervision: Yindong Li.

Writing – original draft: Jinxin Chang.

Writing – review & editing: Yindong Li.

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
