## [Decision Letter · Decision Letter 0]

6 Nov 2025

PONE-D-25-27347Deconstructed Tai Chi Step Training Combined with Conventional Rehabilitation Significantly Improves Lower Limb Function in Brunnstrom Stage III Stroke Patients: A Randomized Controlled TrialPLOS ONE

Dear Dr. Li,

Thank you for submitting your manuscript to PLOS ONE. After careful consideration, we feel that it has merit but does not fully meet PLOS ONE’s publication criteria as it currently stands. Therefore, we invite you to submit a revised version of the manuscript that addresses the points raised during the review process.

We look forward to receiving your revised manuscript.

Kind regards,

Aynollah Naderi

Academic Editor

PLOS ONE

Journal Requirements:

2. We note that you have selected “Clinical Trial” as your article type. PLOS ONE requires that all clinical trials are registered in an appropriate registry (the WHO list of approved registries is at https://www.who.int/clinical-trials-registry-platform/network/primary-registries" https://www.who.int/clinical-trials-registry-platform/network/primary-registries and more information on trial registration is at http://www.icmje.org/about-icmje/faqs/clinical-trials-registration/). Please state the name of the registry and the registration number (e.g. ISRCTN or ClinicalTrials.gov) in the submission data and on the title page of your manuscript. a) Please provide the complete date range for participant recruitment and follow-up in the methods section of your manuscript. b) If you have not yet registered your trial in an appropriate registry, we now require you to do so and will need confirmation of the trial registry number before we can pass your paper to the next stage of review. Please include in the Methods section of your paper your reasons for not registering this study before enrolment of participants started. Please confirm that all related trials are registered by stating: “The authors confirm that all ongoing and related trials for this drug/intervention are registered”. Please see http://journals.plos.org/plosone/s/submission-guidelines#loc-clinical-trials for our policies on clinical trials.

**Additional Editor Comments:**

Dear Author,

The study addresses an important and innovative area of post-stroke rehabilitation by exploring a “deconstructed Tai Chi” protocol specifically tailored to patients at Brunnstrom Stage III. The focus on stage-specific intervention is clinically relevant and offers a potentially low-cost, culturally meaningful approach to neurorehabilitation. The quantitative findings are promising and indicate measurable improvements in lower limb motor function. However, several sections require substantial clarification and enhancement to strengthen methodological rigor, theoretical grounding, and interpretive depth.

Reviewers' comments:

Reviewer's Responses to Questions

**Comments to the Author**

1. Is the manuscript technically sound, and do the data support the conclusions?

Reviewer #1: Partly

Reviewer #2: Yes

Reviewer #3: Yes

Reviewer #4: Yes

Reviewer #5: Yes

2. Has the statistical analysis been performed appropriately and rigorously? 

Reviewer #1: No

Reviewer #2: Yes

Reviewer #3: I Don't Know

Reviewer #4: Yes

Reviewer #5: Yes

3. Have the authors made all data underlying the findings in their manuscript fully available?

Reviewer #1: No

Reviewer #2: Yes

Reviewer #3: Yes

Reviewer #4: Yes

Reviewer #5: Yes

4. Is the manuscript presented in an intelligible fashion and written in standard English?

Reviewer #1: Yes

Reviewer #2: Yes

Reviewer #3: Yes

Reviewer #4: Yes

Reviewer #5: Yes

5. Review Comments to the Author

Reviewer #1: Here is a list of specific comments. Note: line and page numbering in reviews and comments is based on ruler applied in Editorial Manager-generated PDF.

1. Page 5, line 94: I suggest adding a subsection to describe the outcomes of interest. If the sentences in lines 203–209 described the outcomes, creating a subsection would satisfy this comment.

2. Page 6, lines 110–111: I suggest stating when the participant in the E group discontinued the intervention and when the participant in the C group lost to follow-up.

3. Page 6, lines 118–119: The sample size was calculated based on a continuous outcome. In this manuscript, five outcomes were assessed. I suggest including a potential inflated type-1 error as a limitation in the Discussion section.

4. Page 7, lines 133–136: The exclusion was post-randomization and made the analysis per-protocol.

5. Page 11, lines 203–209: Assuming these were the outcomes, I suggest including the timing when the outcomes were assessed.

6. Page 11, line 210: I suggest stating the primary population of analysis (e.g., intention-to-treat, per-protocol, etc.)

7. Page 11, lines 215–217: I suggest revising the sentence to clarify that both t-test and Mann-Whitney U test are for continuous variable but one is parametric (t-test) and the other is non-parametric (Mann-Whitney U test).

8. Page 11, lines 215–216: I suggest clarifying if the between-group comparisons were based on the pre-and-post change or post-intervention measurements.

9. Page 12, Table 1: Baseline characteristics were expected to be balanced due to the randomization. I suggest removing the Statistic and p-value columns. If needed, I suggest reporting standardized mean differences.

10. Page 13, lines 238–240: I suggest not reporting the p-values if they comparisons were not the primary objective of the study. Reporting means and 95% confidence intervals would be sufficient. Please apply this comment throughout the manuscript.

11. Page 14, line 245: A binary outcome was not considered when calculating the required sample size.

Reviewer #2: Dear Author,

You have made great effort in crafting a well written manuscript that aligns with the subject area.

In order to enhance the quality and validity of your work, I recommend you ensuring that all references are thoroughly incorporated and updated to reflect the latest developments in the field.

This will strengthen the foundation of your paper and provide readers with a comprehensive understanding of the topic. I encourage you to review and update your references accordingly.

Well done

Reviewer #3: Dear Authors,

I have completed my review of your manuscript and found it to be a well-conducted and valuable study. The research addresses a pertinent clinical question with a robust methodological approach.

Reviewer #4: Overall Evaluation

The topic is clinically relevant, addressing a significant gap in stage-specific post-stroke rehabilitation. The findings are promising and supported by quantitative evidence. However, several aspects, particularly in methodological transparency, discussion depth, and theoretical contextualization require clarification or strengthening to enhance scientific rigor and reproducibility.

Comments

The concept of “deconstructed Tai Chi” tailored to Brunnstrom Stage III is original and aligns with growing interest in culturally adapted, low-cost rehabilitation methods. However, the degree of novelty should be clarified in relation to prior studies on simplified Tai Chi interventions in stroke recovery (e.g., Li et al., 2018; Zou et al., 2019).

The introduction could better distinguish this protocol from prior modified Tai Chi programs by emphasizing how joint isolation targeting extensor synergy specifically meets the neurophysiological needs of Stage III patients. Add 2–3 sentences in the introduction explaining how deconstructed Tai Chi differs in biomechanical intent and progression from previously reported Tai Chi-based interventions for post-stroke recovery.

The introduction currently introduces stroke rehabilitation but lacks sufficient pathophysiological context linking vascular vulnerability and metabolic risk factors to the need for rehabilitative intervention. Add below phrase with recent literature support as shown to strengthens the scientific justification for this study.

“Previous research has emphasized that cerebrovascular fragility and metabolic risk factors, particularly obesity, play a pivotal role in precipitating fatal brain hemorrhages even in individuals without prior clinical symptoms [Sheikh et al., 2025, https://pmc.ncbi.nlm.nih.gov/articles/PMC11875705/ ]. These findings underscore the importance of preventive and restorative strategies aimed at enhancing neurovascular stability and functional resilience, which are core objectives guiding rehabilitation approaches in stroke and related neurological disorders.”

Methodology

Describe how random allocation was performed (e.g., sealed envelopes, computer-generated sequence) and who managed allocation concealment.

While blinding therapists is difficult, please elaborate on how assessor blinding integrity was verified.

Clarify the reference study used for the effect size (d = 1.2). A citation and calculation example would strengthen credibility.

Report dropout reasons (if any) and whether intention-to-treat (ITT) analysis was conducted.

Include a methodological clarification paragraph specifying randomization control, adherence verification, and bias minimization.

Results

The clinical relevance of the between-group difference in FMA-LE (+5.6 vs. +1.8) should be interpreted in the context of minimal clinically important difference (MCID) for post-stroke lower limb function. Additionally, the use of multiple parametric tests across ROM and FMA variables without correction (e.g., Bonferroni) increases the risk of Type I error. Discuss the clinical meaningfulness of the effect size and specify whether adjustments for multiple comparisons were applied.

Discussion

The discussion appropriately highlights neuroplastic and proprioceptive mechanisms; however, it misses deeper integration with motor control theory and neurorehabilitation frameworks.

The section could benefit from citing functional imaging or EMG-based Tai Chi studies to support claims about cortical reorganization.

The discussion currently emphasizes clinical rehabilitation outcomes but misses integration with recent multidisciplinary evidence linking physical activity to neurobiological recovery mechanisms. Add below phrase to strengthen holistic model where physical therapy not only restores motor function but also enhances neuroplasticity, emotional resilience, and overall brain health.

“Emerging evidence has demonstrated that structured physical therapy and exercise interventions can positively modulate neurobiological pathways, promoting neurogenesis, mood stabilization, and functional recovery in neurological disorders [Nasif et al., 2025, https://doi.org/10.31579/2578-8868/359 ]. These findings highlight the broader rehabilitative potential of physical engagement in restoring neurovascular and psychosocial integrity following brain injury or hemorrhage, thereby supporting integrated rehabilitation frameworks for stroke survivors.”

Limitations and Future Scope

Include potential gender imbalance or age-related variability in recovery. Discuss cultural transferability — can this Tai Chi-based model be applied outside East Asia? Consider proposing longitudinal follow-up or home-based adaptation as a research priority. Add one paragraph proposing multicenter validation and wearable-based outcome tracking (e.g., kinematic or EMG-based monitoring) to enhance reproducibility and global applicability.

Reviewer #5: L29–35 (Abstract): Avoid repetitive “significantly improves” phrasing; maintain scientific neutrality.

L85–93 (Introduction): Add explicit hypotheses.

L118–124: Provide numerical justification for sample size calculation.

L210–220: Include CI and η² effect sizes.

L287–320: In Discussion, cite neuroplasticity mechanisms with updated evidence (e.g., Neurorehabilitation and Neural Repair, 2024).

Figures: Label all axes, specify “mean ± SD”, and mark significant differences properly (p<0.05, p<0.01).

Language: Avoid promotional expressions such as “highly recommended for clinical incorporation.” Instead, use “suggests potential for inclusion in clinical practice pending larger trials.”

Clarify randomization and concealment methods.

Include or re-analyze data with repeated-measures ANOVA or mixed models.

Report 95% CI for all major results.

Explicitly state hypotheses and improve Discussion balance.

Update literature (2024–2025) and moderate the tone of conclusions.

6. PLOS authors have the option to publish the peer review history of their article (what does this mean?). If published, this will include your full peer review and any attached files.

Reviewer #1: No

Reviewer #2: **Yes:** Ebenezer Ad Adams

Reviewer #3: No

Reviewer #4: No

Reviewer #5: No

---

## [Author Response · Author response to Decision Letter 1]

11 Apr 2026

Response to Reviewers

Manuscript ID: PONE-D-25-27347

Title: Effects of Deconstructed Tai Chi Step Training Combined with Conventional Rehabilitation on Lower Limb Function in Brunnstrom Stage III Stroke Patients: A Randomized Controlled Trial

Dear Academic Editor and Reviewers,

We sincerely thank you for your careful evaluation of our manuscript and for the constructive comments and suggestions. We have revised the manuscript extensively in response to the editor’s and reviewers’ recommendations. Major revisions include clarification of the study rationale and stage-specific innovation, improvement of methodological transparency, addition of explicit outcome and statistical analysis subsections, refinement of the Results presentation, strengthening of the Discussion, updating of the reference list, and moderation of the overall tone of the manuscript. The revised manuscript has been carefully checked for clarity and internal consistency.

Below, we provide a point-by-point response.

Note: Locations refer to section numbers in the revised manuscript.

Responses to Journal Requirements and Editorial Comments

Journal Requirement 1

Comment: Please ensure that your manuscript meets PLOS ONE style requirements, including file naming.

Response:

Thank you for this reminder. We have revised the manuscript to better align with PLOS ONE style expectations, including the title, abstract structure, section organization, and figure/table presentation. We will also ensure that the final uploaded files are named and formatted according to the journal requirements at the time of resubmission.

Journal Requirement 2

Comment: Please provide trial registry information and the complete date range for recruitment and follow-up.

Response:

We have revised the manuscript to provide the full name of the registry and the registration number on the title page and in the Methods section. We now state that the trial was registered in the International Traditional Medicine Clinical Trial Registry (ITMCTR; a WHO ICTRP Primary Registry) under registration number ITMCTR2025000972. We have also added the date range for participant recruitment and the date range for intervention and outcome assessment in Section 2.1.

Journal Requirement 3

Comment: Please ensure that all raw data required to replicate the study findings are fully available.

Response:

We appreciate the importance of this requirement. In the current revised manuscript, the Data Availability Statement remains provisional. Before final resubmission, we will update this statement to comply with the PLOS ONE minimal dataset policy and will provide the de-identified underlying data required to reproduce the reported findings, including the values underlying the tables and figures.

Journal Requirement 4

Comment: Please ensure that the corresponding author’s ORCID iD is validated in Editorial Manager.

Response:

Thank you. The corresponding author will complete ORCID validation in Editorial Manager before resubmission.

Journal Requirement 5

Comment: Please include captions for Supporting Information files at the end of the manuscript.

Response:

Thank you for this important reminder. We will add the captions for all Supporting Information files at the end of the manuscript and ensure that the in-text citations are updated accordingly before final resubmission.

Journal Requirement 6

Comment: Please review and evaluate the publications suggested by reviewers and cite them if relevant.

Response:

We have carefully reviewed the suggested literature and incorporated relevant references where they strengthened the rationale, methodological background, and discussion of mechanisms and rehabilitation implications. We did not add suggested references mechanically; instead, we selectively included those that were directly relevant to the scope and focus of the present study.

Additional Editor Comment

Comment: The study is clinically relevant and innovative, but several sections require clarification to strengthen methodological rigor, theoretical grounding, and interpretive depth.

Response:

We appreciate this overall assessment. In response, we revised the Introduction to better define the stage-specific novelty of the deconstructed Tai Chi protocol, expanded the Methods to clarify randomization, concealment, blinding, attrition, and analysis population, revised the Results to improve statistical reporting, and strengthened the Discussion by integrating motor control theory, rehabilitation frameworks, updated references, and a more balanced interpretation of the findings.

Response to Reviewer 1

Comment 1

I suggest adding a subsection to describe the outcomes of interest.

Response:

Thank you for this suggestion. We have added a dedicated subsection, “2.4 Outcome Measures,” which clearly describes the primary and secondary outcomes, the measurement tools, and the timing of assessments.

Location: Section 2.4.

Comment 2

I suggest stating when the participant in the E group discontinued the intervention and when the participant in the C group lost to follow-up.

Response:

We have added the specific time points and reasons. One participant in the experimental group discontinued the intervention at week 4 because of family obligations, and one participant in the control group was lost to follow-up after week 6 and could not be contacted.

Location: Section 2.2.2

Comment 3

The sample size was calculated based on a continuous outcome. In this manuscript, five outcomes were assessed. Please include potential inflated type I error as a limitation.

Response:

We agree. We have addressed this point in two places. In Section 2.5, we now state that no formal adjustment for multiple comparisons was applied and that the secondary ROM analyses should be interpreted cautiously. We also added this issue explicitly to the Limitations section.

Location: Sections 2.5 and 4.4.

Comment 4

The exclusion was post-randomization and made the analysis per-protocol.

Response:

We agree and have clarified this point. The revised manuscript now distinguishes pre-randomization exclusions from post-randomization discontinuations and explicitly states that the final analysis was conducted on a per-protocol basis. We also state that intention-to-treat analysis was not performed.

Location: Sections 2.2.2 and 2.5.

Comment 5

Please include the timing when the outcomes were assessed.

Response:

We have added this information explicitly. The revised manuscript now states that outcome measures were assessed at baseline before randomization and again immediately after completion of the 8-week intervention period.

Location: Section 2.4.

Comment 6

Please state the primary population of analysis (e.g., intention-to-treat, per-protocol, etc.).

Response:

We have now clearly stated that the primary analysis was conducted on a per-protocol basis.

Location: Sections 2.2.2 and 2.5.

Comment 7

Please revise the sentence to clarify that both t-test and Mann–Whitney U test are for continuous variables, but one is parametric and the other is non-parametric.

Response:

We have revised the Statistical Analysis section to clarify that independent t-tests were used for normally distributed continuous variables, whereas Mann–Whitney U tests were used for non-normally distributed variables and for ordinal data where appropriate.

Location: Section 2.5.

Comment 8

Please clarify whether the between-group comparisons were based on pre-and-post change or post-intervention measurements.

Response:

We have explicitly stated that between-group comparisons for continuous outcomes were based on change scores (post-intervention minus baseline).

Location: Section 2.5.

Comment 9

Baseline characteristics were expected to be balanced due to the randomization. Please remove the Statistic and p-value columns.

Response:

We agree. Table 1 has been revised to present baseline characteristics descriptively only. The Statistic and p-value columns were removed, and the corresponding Results text was revised accordingly.

Location: Section 3.1 and Table 1.

Comment 10

Please do not report p-values if those comparisons were not the primary objective of the study. Reporting means and 95% confidence intervals would be sufficient.

Response:

We appreciate this suggestion. We revised the Results section to emphasize descriptive changes, 95% confidence intervals, and effect sizes for the major outcomes. We also reduced overreliance on repeated significance statements, particularly in the presentation of the continuous outcomes.

Location: Sections 3.2–3.4.

Comment 11

A binary outcome was not considered when calculating the required sample size.

Response:

Thank you. We have clarified in Section 2.2.1 that the sample size calculation was based on the primary continuous outcome. We also acknowledge in the Discussion/Limitations that additional outcomes, including Holden classification, were analyzed beyond the outcome used for sample size estimation.

Location: Sections 2.2.1 and 4.4.

Response to Reviewer 2

Comment

Please ensure that all references are thoroughly incorporated and updated to reflect the latest developments in the field.

Response:

Thank you for this helpful recommendation. We reviewed and updated the reference list, particularly in the Introduction and Discussion. The revised manuscript now includes additional recent studies where relevant to strengthen the background, methodological framing, and discussion of mechanisms and rehabilitation implications.

Location: Introduction, Discussion, and References.

Response to Reviewer 3

Comment

The study addresses a pertinent clinical question with a robust methodological approach.

Response:

We sincerely thank the reviewer for this positive and encouraging evaluation of our work.

Response to Reviewer 4

Comment 1

The novelty of “deconstructed Tai Chi” should be clarified in relation to prior studies on simplified Tai Chi interventions in stroke recovery.

Response:

We appreciate this comment and revised the Introduction accordingly. The revised text now distinguishes the present protocol from previously reported modified or simplified Tai Chi interventions by emphasizing that our approach was specifically designed for the motor characteristics of Brunnstrom Stage III, including extensor synergy dominance, limited selective joint control, and impaired weight shifting. We also clarify that the present protocol reorganizes training according to stage-specific biomechanical and neurophysiological demands rather than merely reducing movement complexity.

Location: Introduction.

Comment 2

The Introduction lacks sufficient pathophysiological context.

Response:

Thank you. Rather than adding the suggested wording verbatim, we strengthened the Introduction in a way that remained closely aligned with the focus of our study. Specifically, we added text emphasizing disrupted neuromuscular coordination, impaired postural control, limited weight transfer, and the relevance of targeted intervention during the subacute stage of recovery.

Location: Introduction.

Comment 3

Describe how random allocation was performed and who managed allocation concealment.

Response:

We have revised Section 2.1 to state that random allocation was performed using a computer-generated random sequence with a block size of 4, and that the allocation sequence was concealed in sequentially numbered, opaque, sealed envelopes prepared and managed by an independent researcher not involved in recruitment, intervention delivery, or outcome assessment.

Location: Section 2.1.

Comment 4

Please elaborate on how assessor blinding integrity was verified.

Response:

We expanded Section 2.2.3. The revised manuscript now states that after each evaluation session, assessors were asked to guess group assignment, and that blinding was considered successful if the correct guess rate did not exceed chance level. The observed guess accuracy was 38%, suggesting that assessor blinding was likely maintained throughout the study.

Location: Section 2.2.3.

Comment 5

Clarify the reference study used for the effect size (d = 1.2).

Response:

We clarified in Section 2.2.1 that the sample size calculation was based on the primary continuous outcome and used an assumed effect size of d = 1.2, derived from the between-group difference reported in Zhao et al.

Location: Section 2.2.1.

Comment 6

Report dropout reasons and whether ITT analysis was conducted.

Response:

This information has now been added to Section 2.2.2. We report the timing and reasons for the two post-randomization losses and explicitly state that the final analysis was conducted on a per-protocol basis and that intention-to-treat analysis was not performed.

Location: Section 2.2.2.

Comment 7

Include a methodological clarification paragraph specifying randomization control, adherence verification, and bias minimization.

Response:

We addressed this request by revising Sections 2.1, 2.2.3, and 2.3.4. These revisions clarify random sequence generation and concealment, adherence monitoring using attendance logs, standardized therapist training, scripted intervention delivery, and blinded outcome assessment.

Location: Sections 2.1, 2.2.3, and 2.3.4.

Comment 8

Interpret the clinical relevance of the between-group FMA-LE difference in the context of MCID, and clarify whether multiple-comparison adjustment was applied.

Response:

We appreciate this important point. In the revised manuscript, we added 95% confidence intervals and effect sizes for the major outcomes and discussed the potential clinical relevance of the FMA-LE difference more cautiously in the Discussion. We also clarified in Section 2.5 that no formal adjustment for multiple comparisons was applied and that the secondary ROM analyses should therefore be interpreted cautiously. This issue is also acknowledged in the Limitations section.

Location: Sections 2.5, 4.2, and 4.4.

Comment 9

The Discussion should better integrate motor control theory and neurorehabilitation frameworks.

Response:

Thank you. We substantially revised Section 4.1 to incorporate motor control theory, task-specific repetition, progressive joint dissociation, controlled weight transfer, and a motor relearning framework.

Location: Section 4.1.

Comment 10

The Discussion could benefit from citing functional imaging or EMG-based Tai Chi studies to support claims about cortical reorganization.

Response:

We appreciate this suggestion. While we did not identify a functional imaging study that matched our protocol and patient stage closely enough to cite directly, we strengthened the mechanistic discussion by adding updated EMG-/mechanistic-/neuromodulation-related references and by clarifying that no neurophysiological measures were collected in the present study. This limitation is now explicitly stated.

Location: Sections 4.1 and 4.4.

Comment 11

Please integrate broader evidence linking physical activity to neurobiological recovery.

Response:

We revised the Discussion to broaden the interpretation of the intervention beyond biomechanical practice alone. The revised text now notes that structured exercise may contribute to neurobiological recovery processes relevant to stroke rehabilitation and integrates this idea into the overall interpretation of the findings.

Location: Sections 4.1 and 4.3.

Comment 12

Please expand the limitations and future directions.

Response:

We revised Section 4.4 to address this comment directly. The revised section now includes the small sample size, per-protocol analysis, lack of neurophysiological assessment, possible performance bias, age- and sex-related variability, cultural transferability, home-based adaptation, multicenter validation, and the future use of wearable sensors, gait kinematics, and surface EMG.

Location: Section 4.4.

Response to Reviewer 5

Comment 1

Avoid repetitive “significantly improves” phrasing; maintain scientific neutrality.

Response:

We revised the title, abstract, and conclusion to adopt a more neutral tone. Expressions such as “significantly improves” and other stronger promotional language were replaced with more cautious formulations such as “was associated with greater improvement” and “may represent a practical adjunct.”

Location: Title, Abstract, and Conclusion.

Commen

---

## [Editor Report · Decision Letter 1]

21 Apr 2026

DEffects of Deconstructed Tai Chi Step Training Combined with Conventional Rehabilitation on Lower Limb Function in Brunnstrom Stage III Stroke Patients: A Randomized Controlled Trial

PONE-D-25-27347R1

Dear Dr. Yindong Li,

We’re pleased to inform you that your manuscript has been judged scientifically suitable for publication and will be formally accepted for publication once it meets all outstanding technical requirements.

Kind regards,

Aynollah Naderi

Academic Editor

PLOS One

Additional Editor Comments (optional):

Your comprehensive, transparent, and well-structured responses to all points raised by the reviewers and the editor demonstrate a clear commitment to strengthening methodological rigor, reporting transparency, and balanced scientific interpretation. The implemented revisions—including the addition of a dedicated outcomes subsection, reporting of 95% confidence intervals and effect sizes, appropriate use of repeated-measures ANOVA, clarification of randomization and assessor blinding procedures, removal of p-values from the baseline characteristics table, and moderation of the manuscript’s tone—are fully aligned with CONSORT guidelines and PLOS ONE requirements, and have effectively resolved the reviewers’ primary concerns. Provided that the authors finalize the data availability statement, validate the corresponding author’s ORCID iD, and ensure strict adherence to the journal’s formatting instructions prior to final submission, the manuscript is scientifically and methodologically sound and ready for acceptance.
---

## [Editor Report · Acceptance letter]

PONE-D-25-27347R1

PLOS One

Dear Dr. Li,

I'm pleased to inform you that your manuscript has been deemed suitable for publication in PLOS One. Congratulations! Your manuscript is now being handed over to our production team.

Kind regards,

on behalf of

Dr. Aynollah Naderi

Academic Editor

PLOS One